# In Vivo Efficacy of Curcumin and Curcumin Nanoparticle in *Trypanosoma congolense*, Broden 1904 (Kinetoplastea: Trypanosomatidae)-Infected Mice

**DOI:** 10.3390/pathogens12101227

**Published:** 2023-10-09

**Authors:** Nthatisi Innocentia Molefe-Nyembe, Oluyomi Stephen Adeyemi, Daisuke Kondoh, Kentaro Kato, Noboru Inoue, Keisuke Suganuma

**Affiliations:** 1Department of Zoology and Entomology, University of the Free State, Private Bag X13, Phuthaditjhaba 9866, South Africa; 2National Research Center for Protozoan Diseases, Obihiro University of Agriculture and Veterinary Medicine, Nishi 2-11 Inada, Obihiro 080-8555, Hokkaido, Japan; ircpmi@obihiro.ac.jp (N.I.); k.suganuma@obihiro.ac.jp (K.S.); 3Department of Biochemistry, Medicinal Biochemistry and Toxicology Laboratory, Landmark University, PMB 1001, Ipetu Road, Omu-Aran 251101, Nigeria; oluyomiadeyemi@gmail.com; 4Section of Anatomy and Pathology, Division of Veterinary Sciences, Department of Veterinary Medicine, Obihiro University of Agriculture and Veterinary Medicine, Nishi 2-11 Inada, Obihiro 080-8555, Hokkaido, Japan; kondoh-d@obihiro.ac.jp; 5Laboratory of Sustainable Animal Environmental Systems, Graduate School of Agricultural Sciences, Tohoku University, Sendai 980-8577, Japan; kkato@obihiro.ac.jp

**Keywords:** animal African trypanosomosis, drug discovery, nanomedicine, natural products, phytomedicine

## Abstract

Curcumin (CUR) is known for its wide folkloric effects on various infections; however, its solubility status has remained a hindrance to its bioavailability in the host. This study evaluated the comparative effects of CUR and CUR-nanoparticle in vitro on *T. congolense*, *T. b. brucei*, and *T. evansi.* Additionally, CUR and CUR-nanoparticle anti-*Trypanosoma* efficacy were assessed in vivo against *T. congolense*. All the CUR-nanoparticles were two folds more effective on the *T. congolense* as compared to CUR in vitro, with recorded efficacy of 3.67 ± 0.31; 7.61 ± 1.22; and 6.40 ± 3.07 μM, while the CUR-nanoparticles efficacy was 1.56 ± 0.50; 28.16 ± 9.43 and 13.12 ± 0.13 μM on *T. congolense*, *T. b. brucei*, and *T. evansi*, respectively. Both CUR and CUR-nanoparticles displayed moderate efficacy orally. The efficacy of CUR and CUR-nanoparticles in vivo was influenced by solubility, presence of food, and treatment period. CUR-treated mice were not cured of the infection; however, the survival rate of the orally treated mice was significantly prolonged as compared with intraperitoneal-treated mice. CUR-nanoparticles resulted in significant suppression of parasitemia even though relapsed was observed. In conclusion, CUR and CUR-nanoparticles possess moderate efficacy orally on the trypanosomes as compared to the intraperitoneal treatment.

## 1. Introduction

Animal African trypanosomosis (AAT) is a debilitating disease of animals caused by parasites of various species of blood- and tissue-dwelling protozoan of the genus *Trypanosoma*, Broden 1904 (Kinetoplastea: Trypanosomatidae) [1]. *Trypanosoma congolense* and *T. brucei brucei* are the main causative agents of nagana in cattle, sheep, goats, and dogs, while *T. evansi* causes surra in a wide variety of hosts such as the equines, camels, goats, sheep, cattle, dogs, tigers and buffaloes [1,2]. *Trypanosoma congolense* and *T. b. brucei* are cyclically transmitted by tsetse flies of the genus *Glossina*, whilst *T. evansi* is mechanically transmitted by blood-sucking flies of the genus *Tabanus* and *Stomoxys* [3].

Trypanosomosis is widely distributed in sub-Saharan Africa, where approximately 36 countries are infested with the presence of the vector and the disease. Most of the infested countries are underdeveloped, poor, heavily indebted, and have food deficits due to the economic burden of the disease [4]. Animals living in endemic areas are less productive in terms of meat, milk, and wool production, as well as draft power, in comparison to the animals in non-endemic areas. Direct and indirect loss due to trypanosomosis is estimated at US $500 million and US $5 billion per annum, respectively [1,4].

Various methods have been implemented to control AAT, such as vector control, reducing the proximity of livestock to the wild reservoir hosts, breeding of trypanotolerant livestock, and the usage of trypanocidal drugs. The main hurdle with the treatment of nagana is the continuous development of resistance against their usage, which could result in a wide outbreak of the disease [4]. The currently available drugs for the treatment of nagana, particularly in Africa, include diminazene aceturate (DA), isometamidium bromide/chloride, homidium bromide/chloride, and the Nifurtimox-eflornithine combination therapy (NECT). These drugs all differ in their mode of action toward the parasite; however, most of them, such as the homidium and diamine group, are known to inhibit the topoisomerase II pathway and the kinetoplasmatic DNA biosynthesis, respectively [2,3,4].

The main challenges towards the sole dependence on these available drugs for the treatment of nagana include the documented toxicity status, parenteral administration (intravenous and intramuscular), which signals the need for a professional technician for the administration of the drug, as well as the prolonged withdrawal period ranging between 21 to 23 days [4]. Most importantly, most of the trypanocidal drugs such as the D,L-alpha-difluoromethylornithine (DFMO), DA, pentamidine, suramin, and melarsoprol are structurally related, which has overtime exacerbated cross-resistance from one drug to the other [1,2,3,4]. Therefore, the current study aimed to test CUR and CUR-nanoparticle for their antitrypanosomal effects in vitro and in vivo in *Trypanosoma-congolense*-infected mice.

Curcumin (CUR) 1,7-bis(4-hydroxy-3-methoxyphenyl)-1,6-heptadiene-3,5-dione (370.4 g/mol) is a natural yellow polyphenol compound extracted from the rhizome of *Curcuma longa* plant. It is commonly used as a food additive, a principal ingredient in foodstuff. Different nations utilize CUR in various ways; for instance, CUR is served as tea in Japan, cosmetics in Thailand, and colorant in Korea [5]. CUR possesses an array of biological and pharmacological effects such as anti-inflammatory, anti-malaria, antimicrobial, anticancer, and anti-diabetic effects and the potential treatment of hypertension, liver injury, and Alzheimer’s disease [6,7,8,9,10]. According to Witkin and Li [7], CUR expresses its diverse biological action on the cells using three major systems, namely, inflammatory, cell death pathways linked to nuclear factor ᴋB (NF-ᴋB), and the oxidative stress system. Additionally, CUR possesses a beneficial action of scavenging reactive oxygen species (ROS) and, therefore, combating oxidative stress caused by either pathogens or metabolic processes. Moreover, the efficacy of CUR may be due to its ability to impact cell adhesion and neurogenesis, which has thus far resulted in documented cell arrest, apoptosis by caspase induction, and inhibition of cell proliferation and metastasis of cancer cells [9,11]. Furthermore, many clinical trials involving CUR have been conducted globally on several inflammatory diseases and cancer [12].

CUR and turmeric products have been characterized as safe by the Food and Drug Administration (FDA) in the United States of America, the Natural Health Products Directorate of Canada, and the joint Food and Agriculture Organization (FAO)/WHO Expert Committee and have been approved as “Generally Recognized As Safe” (GRAS) [5,10]. CUR is a relatively inert compound and, thus far, has not appeared toxic in either animals or humans. Supporting studies of Witkin and Li [7] and Moghadamtosi et al. [13] have, respectively, reported that CUR is tolerable at concentrations of 8 g/day and 12 g/day. Regardless of all the documented efficacy, CUR is still not yet a pharmaceutically available drug due to its poor pharmacokinetics and/or physicochemical properties. The hydrophobic nature (solubility: 20 mg/mL in water) of CUR constitutes a primary constriction, resulting in low bioavailability because of poor gastrointestinal tract absorption. Furthermore, the compound lacks stability in an alkaline medium, even though it is quite stable in an acidic medium, with poor bioavailability. Lastly, CUR is rapidly hydrolyzed in an alkaline medium, followed by molecular fragmentation [9,14,15,16].

To improve the pharmacokinetics and/or physicochemical properties of CUR, researchers have embraced the application of nanotechnology. This initiative has positively affected the pharmacokinetics of CUR, with reported improvement in the aqueous solubility, bioavailability, alkaline stability, prolonged circulation, better permeability, and biocompatibility of the compound [17,18]. In the biomedical field, the use of nanotechnology, commonly referred to as nanomedicine, is constantly growing. Nanomedicine is a branch of medicine that utilizes particles smaller than 100 nm in size for the treatment of various diseases and infections. These particles are useful in a diverse range of applications. In the pharmaceutical field, nanoparticles are seen as a vital means of overcoming challenges facing hydrophobic agents, utilized in the form of a drug delivery system. The nanoparticles can be in various forms, such as liposomes, micelles, and encapsulated nanoparticles, phospholipid complexes [11].

A tremendous improvement in the bioavailability of several compounds has been achieved using nanotechnology, which, amongst others, are trypanocidal compounds, namely, diminazene aceturate (DA), pentamidine, and suramin [19,20]. Above all, the utilization of nanotechnology can enhance the efficacy of the drugs as compared to the free compounds [21,22]. Several studies have demonstrated the trypanocidal effect of curcumin on *Trypanosoma evansi*, *T. b. brucei*, and *T. cruzi* [23,24,25].

In this study, we assessed the comparative in vitro anti-*Trypanosoma* efficacy of CUR and CUR-nanoparticles against *T. congolense*, *T. b. brucei*, and *T. evansi* bloodstream forms, as well as in vivo evaluation in *T. congolense*-infected mice model. Additionally, we determined the cytotoxicity of CUR on mammalian cell lines.

## 2. Materials and Methods

### 2.1. Curcumin Compound

CUR [1,7-bis(4-hydroxy-3-methoxyphenyl)-1,6-hep-tadiene-3,5-dione] compound was purchased from Sigma Aldrich, Tokyo, Japan.

### 2.2. Preparation of Nanoparticles

The CUR-nanoparticles were prepared using three various methods, as described below.

#### 2.2.1. Antisolvent Precipitation with a Syringe Pump (ASPS)

The solution of CUR was prepared in ethanol at a predetermined concentration of 20 mg/mL. The syringe was filled with 10 mL of the prepared solution and secured onto a syringe pump. The drug solution was quickly injected at a fixed flow rate of 2–10 mL/min into a deionized water that acted as the antisolvent of a definite volume on a magnetic stirrer at 1000 rpm. The ratio of ethanol to water used was 1:10 (*v*/*v*), corresponding to a volume of 100 mL of water to 10 mL of the CUR solution. The CUR-nanoparticles formed were filtered and vacuum dried (Taitec VD-500R freeze dryer, Taitec corporation, Tokyo, Japan) [26].

#### 2.2.2. Evaporative Precipitation of Nanosuspension (EPN)

The solution of CUR was prepared in ethanol (Sigma-Aldrich, Tokyo, Japan), and then a nanosuspension was formed by quickly adding hexane (Wako Pure Chemical Ltd., Osaka, Japan) that acted as an antisolvent. CUR particles in the nanosuspension were obtained using quick evaporation of the solvent and antisolvent under a vacuum evaporator to obtain dry particles [26].

#### 2.2.3. Wet-Milling Method (WN)

A stock of 20 mg/mL (54 µM) CUR solution was prepared by dissolving CUR powder in 10 mL dichloromethane (Sigma-Aldrich, Tokyo, Japan). The solution was injected into a syringe and secured onto a syringe pump. The solution was added dropwise into deionized boiling water under ultrasonication conditions with ultrasonic power (Branson, 3510 Yamato, Tokyo, Japan) at a frequency of 50 kHz. The solution was sonicated for 30 min, and then the mixture was stirred at 800 rpm for 20 min until the orange-colored precipitate was obtained. Thereafter, the solution was vacuum-dried to obtain the powdered CUR nanoparticles [27].

#### 2.2.4. Preliminary Detection of CUR-Nanoparticle Using UV-Visible Spectroscopy

The preliminary detection of synthesized CUR-nanoparticles was carried out using a UV-visible spectrophotometer (ThermoFischer Scientific, Tokyo, Japan), scanning the absorbance spectra in the range of 200–800 nm wavelength.

#### 2.2.5. Particle Size and Shape Analysis

The synthesized CUR nanoparticles were subjected to TEM analysis to determine their size and shape. The nanoparticles were dissolved in deionized water to prepare a 10 mg/mL solution. The solution was added to the carbon grid and left overnight at room temperature. Once dried, the samples were then analyzed at Hanaichi Ultrastucture Research Institute, Aichi, Japan (Camera: XR16, Exposure (MS): 3200; Gain: 1, Bin: 1; Direct magnification: 120,000×, Gamma:1, No sharping, No contrast). Additionally, ImageJ software 1.51t version (NIH, Bethesda, MA, USA) was used to calculate the average particle size of the nanoparticles.

### 2.3. Trypanosome Cultures

The *T. congolense* (IL3000); *T. evansi* (Tansui); and *T. b. brucei* (GUTat 3.1) bloodstream forms (BSF) were propagated at 33 °C for *T. congolense* and 37 °C for *T. evansi* and *T. b. brucei* using the HMI-9 medium [28].

#### In Vitro Anti-Trypanosoma Assay

The bloodstream form (BSF) of *T. congolense*, *T. b. brucei*, and *T. evansi* in the log phase were seeded at 1 × 10^5^, 1 × 10^4^, and 2 × 10^4^ cells/mL, respectively, in a Nunc™ 96-well optical bottom plate (ThermoFisher Scientific) and exposed to various concentrations of CUR and CUR-nanoparticles (3.67–50.52 μM; Tokyo Chemical Industry Co., Ltd., Tokyo, Japan). The plates were incubated for 72 h. Subsequently, 25 μL of CellTiter-Glo™ Luminescent Cell Viability Assay reagent (Promega Japan, Tokyo, Japan) was added to evaluate intracellular ATP concentration. The plate was shaken for 2 min (500 shakes/min) using an MS3 basic plate shaker (IKA^®^ Japan K.K., Osaka, Japan) to facilitate cell lysis and the release of intracellular ATP. The plates were further incubated for 10 min at room temperature and were subsequently read using a GloMax^®^-Multi+ Detection System plate reader (Promega, Tokyo, Japan) [29]. The experiments were conducted in triplicate.

### 2.4. Mammalian Cell Line Cultures

The Madin-Darby bovine kidney (MDBK cell, NBL-1 strain: JCRB cell bank) and mouse embryonic fibroblast, NIH 3T3 cell lines (courtesy of Professor Makoto Igarashi of the National Research Center for Protozoan Diseases) were cultured in Minimum Essential Medium Eagle (MEM) suspended with 10% HI-FBS at 37 °C in an incubator under 5% CO_2_. The cells were maintained by replacing the medium with fresh medium 2 days before the cells became confluent.

#### In Vitro Cytotoxicity Tests

Mammalian cell lines (MDBK and NIH 3 T3) in log phase were seeded separately at a concentration of 1 × 10^5^ cells/mL in a 96-well microtiter plate (ThermoFisher Scientific) and exposed to concentrations ranging from 0.86 to 67.5 µM (8 serial dilutions) of CUR and CUR-nanoparticles. The cell viability was determined using a CCK-8 (Dojindo Laboratories, Kumamoto, Japan) assay 72 h post-incubation. The surviving cells were counted using an ELISA reader (MTP 500, Corona Electric, Hitachinaka-shi, Japan) according to the amount of formazan that formed at the absorbance of 450 nm. The experiments were conducted in triplicate [29].

The selectivity index was calculated to allow for the possible identification of compounds with high efficacy and selective toxicity.
Selectivity index = minimum toxic concentration (μM)/minimum inhibitory concentration (μM)
where the minimum toxic concentration is the compound concentration that inhibited 50% of the cell growth (LC_50_), while the minimum inhibitory concentration was the concentration that inhibited the proliferation of the parasite by 50% (IC_50_), selectivity indices for each cell line were calculated against each trypanosome species.

### 2.5. Animal Experiments

Healthy female BALB/c mice (CLEA Japan Inc., Tokyo, Japan) weighing 20–30 g were kept in the animal house of the National Research Center for Protozoan Diseases of Obihiro University of Agriculture and Veterinary Medicine, Japan. The animals were acclimatized in plastic cages in an air-conditioned environment and were maintained at 25 ± 2 °C with 60 ± 10% relative humidity under a 12 h light and dark cycle for 1 week before commencing the experiments. All the animals had *ad libitum* access to normal mice feed and water. The experiment was approved by the animal ethics committee of Obihiro University of Agriculture and Veterinary Medicine, Japan (approval nos. 28-129 and 28-169).

#### 2.5.1. Histopathological Analysis

The heart, liver, kidneys, and spleen tissues were collected for histological studies from healthy mice that received CUR and CUR-nano in corn oil for 28 consecutive days. The tissues were washed in normal saline and fixed immediately in 10% formalin overnight, dehydrated with isopropanol alcohol, embedded in paraffin, cut into 4–5 µm thick sections, and stained with hematoxylin-eosin (H and E) dye for photomicroscopic observation. The microscopic features of treated groups were viewed under the light microscope (Nikon Eclipse E100 Company, Shinagawa, Japan) and compared with the control group.

#### 2.5.2. The In Vivo Anti-Trypanosoma Effects of CUR and CUR-Nanoparticles on *Trypanosoma Congolense*-Infected Mice

AOral administration

The virulent *T. congolense* IL3000 strain was propagated in mice and passaged twice in mice before the experiment. The mice were intraperitoneally injected with 100 µL mixture of *T. congolense* (5 × 10^3^ parasites/mouse) and phosphate-buffered saline with 1% glucose (PSG). The mice were randomly divided into seven groups consisting of five mice as follows: Group I (control group), the mice were infected but not treated; group II (positive drug control group), the mice were infected and treated with diminazene aceturate (DA—3.5 mg/kg, intraperitoneally) (Sigma Aldrich, Tokyo, Japan); Groups III, IV, V, VI and VII (the test groups), the mice were infected and orally treated using a feeding needle with 50, 100, 200, 300 and 400 mg/kg CUR and ASPS nanoparticles, in a 200 μL inoculum, respectively. Treatment was initiated at 48 h post-infection and was maintained for seven consecutive days. The treatments were freshly prepared each day. The surviving mice were observed for 90 days, while the others were observed until death. Each day, the parasitemia was evaluated, and the effects of treatment were monitored using wet blood smears. Each slide was prepared with fresh blood collected from the tail vein and viewed under a light microscope (Nikon Eclipse E100 Company, Shinagawa, Japan; magnification: 400×). The experiments were conducted in duplicate [29].

BIntraperitoneal administration

The virulent *T. congolense* IL3000 strain was propagated in mice and used for infection. The parasites were passaged twice in mice before the experiment. The mice were intraperitoneally infected with 100 µL of *T. congolense* (5 × 10^3^ parasites/mouse) inoculated with PSG. The mice were randomly divided into nine groups consisting of 5 mice as follows: Group I (control group), the mice were infected but not treated; Group II (positive drug control group), the mice were infected and treated with diminazene aceturate (DA) (3.5 mg/kg, intraperitoneally); Groups III, IV, V, VI, VII, VII, IX (the test groups), the mice were infected and intraperitoneally treated with 50, 75, 100, 150 and 200 mg/kg CUR and CUR-nanoparticles, respectively. Treatment was initiated at 48 h post-infection and was maintained for 7 consecutive days. The treatments were freshly prepared each day. The surviving mice were observed for 90 days, while the others were observed until death. Each day, the parasitemia was evaluated, and the effects of treatment were monitored using wet blood smears [29]. Each slide was prepared with fresh blood collected from the tail vein and viewed under a light microscope (Nikon Eclipse E100 Company, Shinagawa, Japan; magnification: 400×). The experiments were conducted in duplicate.

### 2.6. Statistical Analysis

The results were expressed as the mean ± standard deviation (S.D.) for the number of repeated trials indicated in each experiment. The statistical analyses were conducted in the acute phase of infection. The *t*-test was used for intergroup comparisons between the treated and non-treated groups. The survival curves were constructed using the Kaplan–Meier method, and the curves were compared using a log-rank test. All the data were compiled using the GraphPad Prism Software program (version 5.0, GraphPad Software Inc., La Jolla, CA, USA). *p* values of <0.05 were considered to indicate statistical significance.

## 3. Results

Preliminarily, the CUR-nanoparticles were detected using UV-visible spectroscopy scanned in the range of 200 to 800 nm. The typical CUR absorption was obtained with an observed peak at 419 nm (Figure 1A). TEM imaging of aqueous dispersion showed that the average particle size of the CUR-nanoparticle was 120 nm (Figure 1B). Furthermore, the particle shape varied between an oval and a spherical shape.

### 3.1. In Vitro Anti-Trypanosoma and Cytotoxic Effects

The efficacy of CUR-nanoparticles was 2-fold higher than the free CUR. The CUR and CUR-nanoparticles showed higher efficacy against *T. congolense* than against *T. b. brucei* and *T. evansi*, regardless of the preparation method used. Nonetheless, the ASPS-prepared CUR-nanoparticles showed a remarkable efficacy on all the trypanosome strains in vitro, with less cytotoxicity recorded on the mammalian cells, even though the selectivity index was not superior to that of the WM-prepared CUR-nanoparticles. There were low cytotoxicity effects observed for CUR and CUR-nanoparticles with the recorded LC_50_ values greater than 48.86 µM on the MDBK cells. A moderate cytotoxicity effect was observed when the NIH 3T3 cells were treated with the pure CUR compound (Table 1).

### 3.2. In Vivo Tests

#### 3.2.1. Histopathological Analysis

Histological sections of the duodenum and ileum in female mice exposed to pure CUR treatment, in the absence of CUR-nano, showed a dose-dependent atrophy of the villi and the thinning of the mucosa propria (Figure 2 and Figure 3) as compared to the vehicle control group. No organ alterations were observed in CUR-nano-treated mice at all concentrations.

#### 3.2.2. Oral Administration

ACUR

Parasitemia levels were significantly reduced in all the treated groups on days 4, 5, 6, and 7 as compared to the control group. A low parasitemia was obtained and maintained for some days in the treated groups, whilst that of the control group was observed on day 8, resulting in the death of the mice. Parasites were not cleared completely from the blood circulation but significantly suppressed and allowed a longer survival of the mice in all the treated groups. None of the mice was cured in all the groups except those in the DA-treated group. At 400 mg/kg, on day 14, the mice showed the ability to withstand high infection, illustrated as the highest peak wave of the whole study (Figure 4).

There was a significant difference in the survival of the mice in the treated groups (*p* < 0.0001) even though none of the mice survived in the CUR-treated groups. The mice did not survive the infection; however, the survival of the mice was prolonged to 16, 17, and 20 dpi, while those in the control group died on day 9 (Figure 5).

BCUR-nanoparticles

The oral treatment of the mice with ASPS-synthesized CUR-nanoparticles resulted in a significant decrease and suppression of the parasitemia in the infected mice between days 5 and 7, in comparison to the control group. As shown in Figure 5, the parasitemia in the control group increased steadily until all the mice died. The treated group displayed the first parasitemia wave on day 6, followed by a decrease in parasitemia. The period between days 7 and 10 showed suppressed parasitemia in all the treated groups. The reappearance of the parasites in the circulation was followed by a rapid increase in parasitemia of the treated group, which led to a total death of the mice in groups 100 and 300 mg/kg, while few in the 50 and 400 mg/kg groups were declared cured from the infection (Figure 6).

There was a significant prolonged survival rate of the mice of *p* < 0.05 at 100, 200, and 300 mg/kg, *p* < 0.01 at 400 mg/kg, and *p* < 0.001 at 50 mg/kg as compared to the control group. On day 9, all the mice in the control group died, while those in 100, 200, and 300 mg/kg died on days 20, 16, and 18, respectively. A total of 10% of mice in the 50 and 400 mg/kg survived until the termination of the study, 90 days post-infection. None of the mice in the DA group died (Figure 7).

#### 3.2.3. Intraperitoneal Administration

The intraperitoneal treatment of *T. congolense*-infected mice showed no efficacy against the infection. Neither mice treated with free curcumin, nor the CUR-nanoparticles were declared cured (Figure 8A and Figure 9A). Mice in both treatment groups died within 8 days. There was no significant difference in the survival of the mice in the treated group in comparison to the non-treated control group (Figure 8B and Figure 9B).

## 4. Discussion

In this study, we synthesized CUR-nanoparticles as a means of improving the efficacy of the compound on trypanosomes. Synthesized CUR-nanoparticles’ average size was 120 nm, which was bigger than the 2–40 nm and 92–110 nm range reported by Bhawana et al. [30] and Pandit et al. [27], but lower if compared to the average size of 330 nm reported by Kakran et al. [26].

In the current study, the comparative anti-*Trypanosoma* effect of CUR and CUR-nanoparticles was tested on three animal trypanosome species, *T. congolense*, *T. b. brucei*, and *T. evansi*, in an in vitro setup. The CUR and CUR-nanoparticles showed higher efficacy on *T. congolense* as compared to *T. b. brucei* and *T. evansi*. Phylogenetically, it has been shown that *T. b. brucei* and *T. evansi* are more closely related than *T. congolense*, which forms a distinct clade together [31]; therefore, the observed distinct efficacy is likely a genetically based object of these three species of animal trypanosomes. The efficacy of CUR against *T. b. brucei* recorded in the study by Nose et al. [23], 4.77 ± 0.91 μM, was slightly lower than the IC_50_ of 7.61 ± 1.22 μg/mL recorded in the current study.

The NIH 3T3 cells were more susceptible to the pure CUR compound as compared to the MDBK, while the cytotoxicity of the CUR-nanoparticles did not differ appreciably between the two mammalian cell lines. The study by Xia et al. [32] reported the susceptibility of mouse-derived cell lines to various compounds; however, it has been documented that cells from the same species of origin and tissue may differ in susceptibility to toxicants. The cytotoxicity of free CUR compound on the NIH 3T3 cells agreed with the studies by Jiang et al. [33], which reported the ability of the compound to induce cellular apoptosis.

The poor pharmacokinetics of CUR limit its prospects and efficacy for biomedical applications. The low aqueous solubility status of curcumin results in poor absorption and, therefore, limited bioavailability [34]. Even though CUR is considerably safe, there have been reports of mild diarrhea, nausea, and skin irritation [6,35]; for this reason, the current study ventured to monitor the extent of diarrhea induced by CUR in healthy mice by providing CUR orally for 28 days in the healthy mice. From the histopathology images (Figure 1 and Figure 2), there were no adverse changes in the morphology of the small intestines except the atrophy of the villi and the thinning of the mucosa propria. During the experiments, diarrhea was observed visually with CUR-colored feces and urine. The presence of atrophy of the villi and the thinning of the mucosa propria demonstrated a dose-dependent induction of diarrhea in mice. Higher concentrations were tested as a measure of follow-up studies in either lower or similar concentrations. The mechanism of diarrhea induced by CUR is not yet understood; however, Kumar et al. [15] reported severe diarrhea that revealed duodenal villous atrophy in patients treated with mycophenolate motifel. Nonetheless, due to the high concentrations trialed for the histopathological analysis, the histological changes in the ileum and duodenum were not significant to combat further animal experiments. According to Hewlings and Kalman [5], curcumin records of diarrhea have been reported entirely due to the lack of bioavailability and absorption. This is despite the documented safety status of CUR reported to range between 500 and 1200 mg/day to experimental organisms for a period of one to four months. These further underscores strategies to improve the physico-chemical properties of CUR, among which is nanosizing.

For the *T. congolense*-infected mice, a mild CUR efficacy was observed in the orally treated mice as compared to the intraperitoneal administration. The efficacy of the oral administration of CUR and CUR-nanoparticles was not significantly different from each other as was anticipated. The efficacy of both compounds might have been affected by various factors, such as the solvent, the drug given with food, and the initial period of the treatment. The regimens of CUR-nanoparticles dissolved in corn oil possessed a better efficacy than the samples dissolved in water. However, the nanosize of the CUR-nanoparticles seems to improve its solubility in water. Previous reports have shown that nanoparticles of higher sizes have the tendency to remain hydrophobic [36]. Furthermore, nanoparticles of larger sizes are prone to coalesce within the cell-membrane-bound vesicle, which means that they are at risk of being phagocytized [36]. Based on this hypothesis, the CUR-nanoparticles produced in this study could have been insignificantly bigger, for they visually dissolved in water. Alternatively, for better results, CUR-nanoparticles should be loaded in lipid core nanocapsules, according to Gressler et al. [24]. More so, in the present study, dissolving CUR-nanoparticles in corn oil seems to improve bioactivity compared with that of water.

CUR-nanoparticles showed higher efficacy in vitro as compared to the free curcumin, which may illustrate a better absorption of the compound in the parasites. The in vivo tests were conducted on *T. congolense* based on the preliminary in vitro tests as well as the selectivity of the compound. However, in the in vivo experiments, the efficacy of CUR-nanoparticles did not give superior effects, producing short-lived efficacy resulting in infection relapses. Moreno et al. [37] stated that the in vitro efficacy does not allow the prediction of therapeutic outcomes in the hosts. This concept is challenging the scientific world, and it may be associated with different modes of action of the active compound in vitro and in the host system [38]. Additionally, the host-pathogen factors may influence the efficacy of the compound in vivo as compared to the medium-cultured parasites tested in vitro. Parasites encounter different environments in the culture medium and the blood containing all the blood cells and other cell types, which may be potentially responsible for the efficacy of the drugs [39]. There was no efficacy whatsoever in the intraperitoneally treated mice, whereby all mice died within 8 days post-infection. The solubility of CUR in organic solvents, such as DMSO, ethanol, and acetone, poses an important hurdle for parenteral administration; there are, therefore, few solvents that are completely safe for parenteral administration. DMSO has widely been used as a powerful solubility tool for lipophilic compounds [30]. DMSO is preferred over the others because of its ability to increase membrane solubility; however, according to previous reports, it is not completely safe when administered directly into the system [40,41], which places the orally effective compounds and solvents in demand.

According to existing studies, various factors affect the effectiveness of nanoparticles [24,36]. Nanoparticles produced at the same time tend to vary in shape and size; hence, sizing of the nanoparticles is vital. As mentioned above, the size of the nanoparticles has the potential to determine the uptake of the compound by macrophages, or they coalesce in the cell membrane, where they will be exposed to the immune defense of the cell. Particle size and surface charge are also responsible for the efficacy of the compound, as well as determining the pathway of the cellular uptake of the particles. The smaller the size of the particle, the better the cellular uptake [24,36]. The spherical and non-spherical particles alike showed good uptake and internalization at one point or the other, depending on the pathogen in question.

The toxicity effects of liposomes were documented in monkeys, while pentamidine encapsulated in polyhexylcyanoacrylate carriers was toxic to mice [42]. The toxicity of nanoparticles can be modulated and regulated based on the type of the study. Nonetheless, the current study only demonstrated the safety of nanoparticles in vitro. The efficacy of CUR-nanoparticles could be improved by changing the carrier of the compound; perhaps lipid-based carriers could be more effective.

In conclusion, oral treatment of CUR-nanoparticle possessed slightly moderate efficacy in vivo, which makes CUR and its nanoparticles a potential candidate for future studies. Potential further studies include the prolonged treatment period of infected mice with CUR and CUR-nano, combination treatment with available trypanocidal drugs, and the determination of the mode of action of the compounds in vitro and in vivo. There is a conspicuous suppression of parasitemia as well as the prolongation of survival of the *T. congolense*-infected mice treatment orally with CUR and CUR-nano. Nonetheless, the efficacy observed in vivo did not imitate the good efficacy observed in the in vitro setting of the study.

## Figures and Tables

**Figure 1 pathogens-12-01227-f001:**
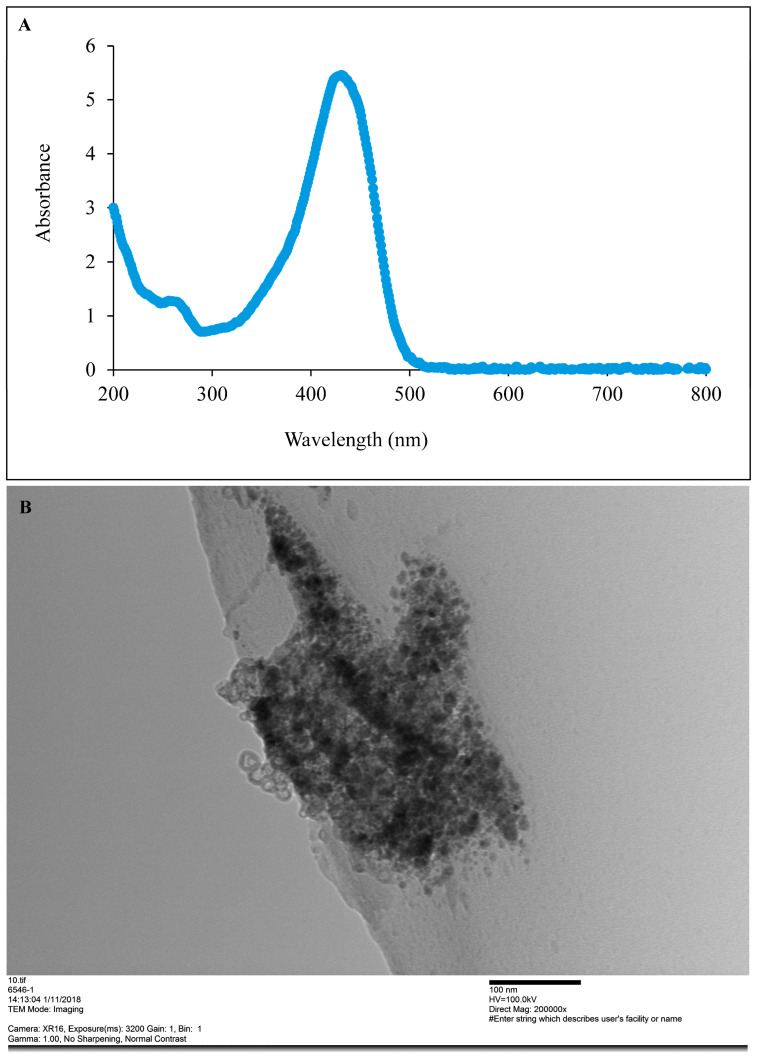
(**A**) UV-VIS spectra, and (**B**) TEM imaging of aqueous dispersion of the synthesized CUR-nanoparticles.

**Figure 2 pathogens-12-01227-f002:**
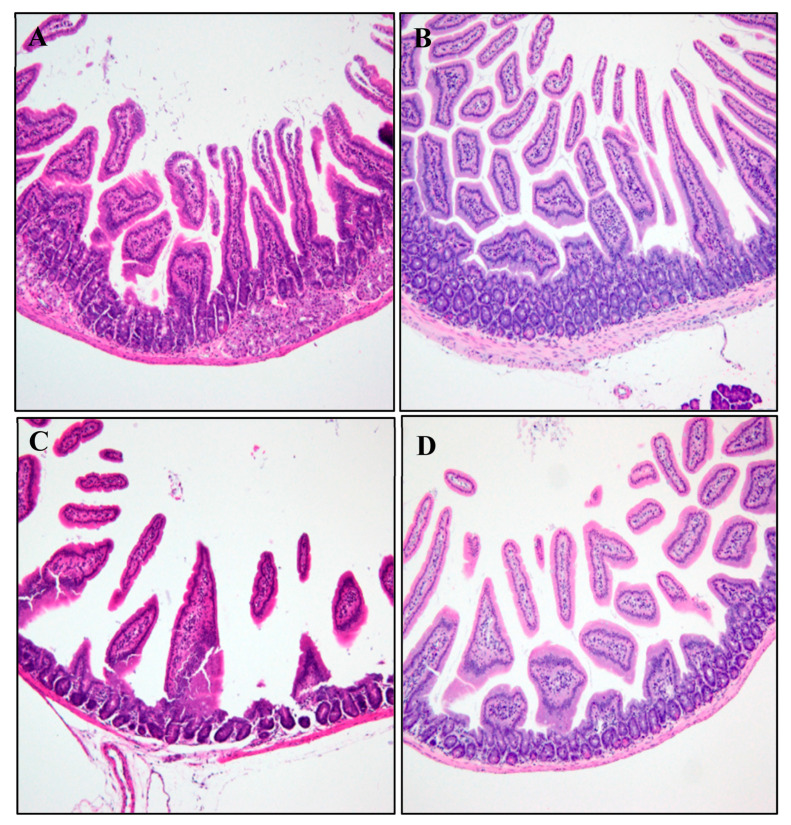
Histology of the duodenum of the control mice and those exposed to various doses of CUR treatment for 28 days: (**A**) control; (**B**) 100 mg/kg; (**C**) 500 mg/kg, and (**D**) 1000 mg/kg.

**Figure 3 pathogens-12-01227-f003:**
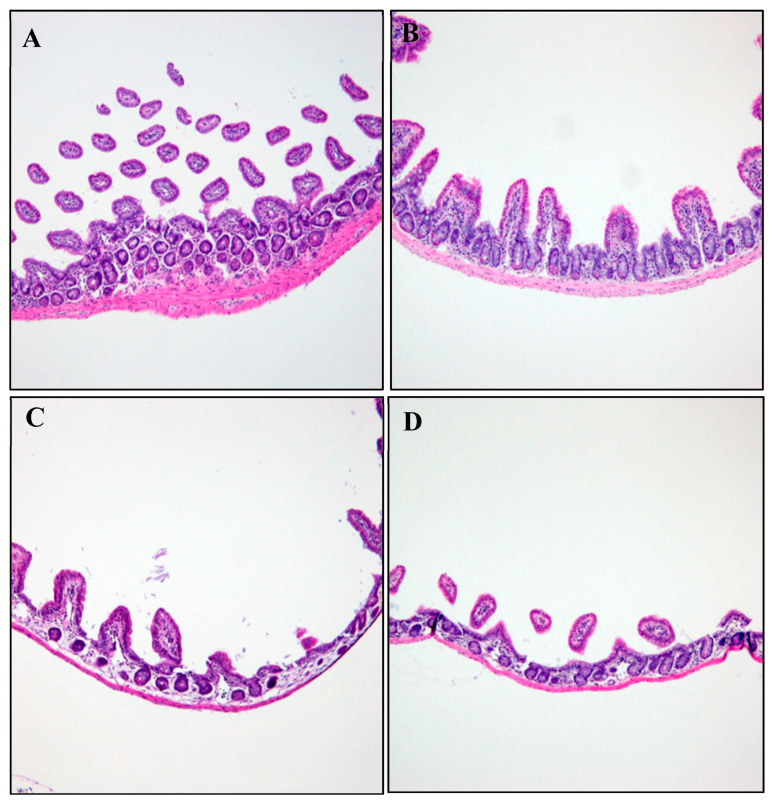
Histology of the ileum of the control mice and those exposed to various doses of CUR treatment for 28 days: (**A**) control; (**B**) 100 mg/kg; (**C**) 500 mg/kg; and (**D**) 1000 mg/kg.

**Figure 4 pathogens-12-01227-f004:**
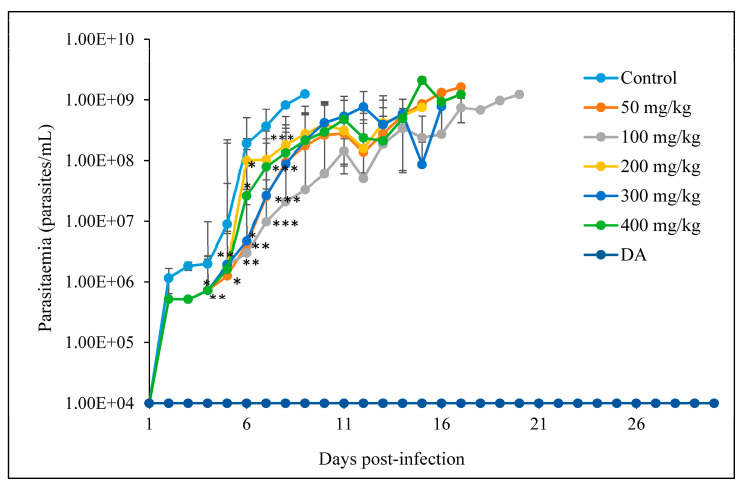
The evaluation of parasitemia in mice infected with *T. congolense* and orally treated with different concentrations of CUR for 7 days. * *p* < 0.05; ** *p* < 0.001; and *** *p* < 0.0001. The data are expressed as the mean ± S.D. The 1 × 10^4^ represents parasitemia below the detection levels.

**Figure 5 pathogens-12-01227-f005:**
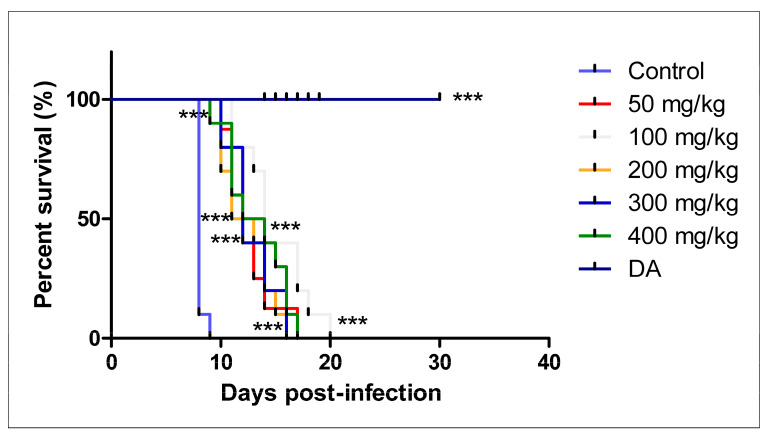
Survival curves of mice infected with *T. congolense* and orally treated with different concentrations of CUR. The survival rate was significantly different from that of the control group (*n* = 10), *** *p* < 0.0001 (Log-rank test).

**Figure 6 pathogens-12-01227-f006:**
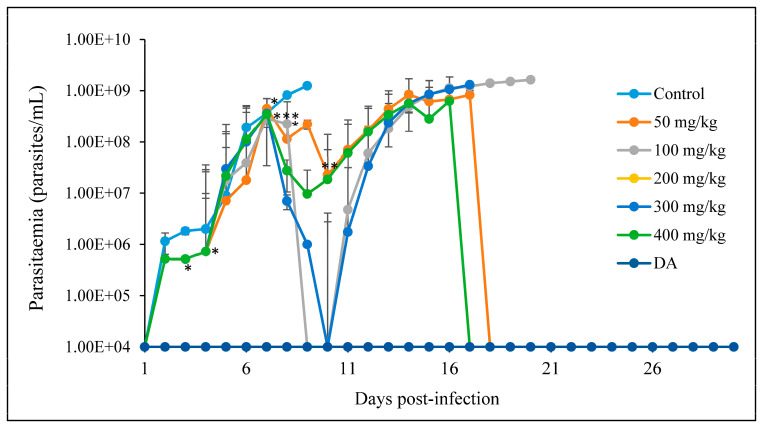
The evaluation of parasitemia in mice infected with *T. congolense* and orally treated with different concentrations of CUR-nanoparticle for 7 days. * *p* < 0.05; ** *p* < 0.001 and *** *p* < 0.0001. The data are expressed as the mean ± S.D. The 1 × 10^4^ represents parasitemia below the detection levels.

**Figure 7 pathogens-12-01227-f007:**
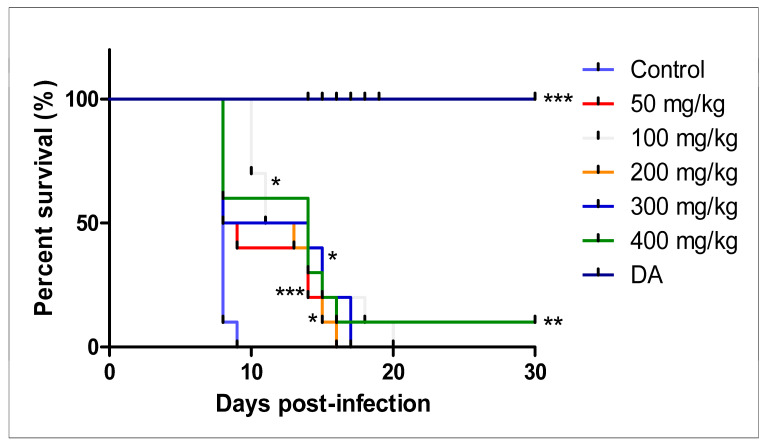
Survival curves of mice infected with *T. congolense* and orally treated with different concentrations of CUR-nanoparticle. The survival rate was significantly different from that of the control group (*n* = 10), * *p* < 0.05, ** *p* < 0.001, *** *p* < 0.0001 (Log-rank test).

**Figure 8 pathogens-12-01227-f008:**
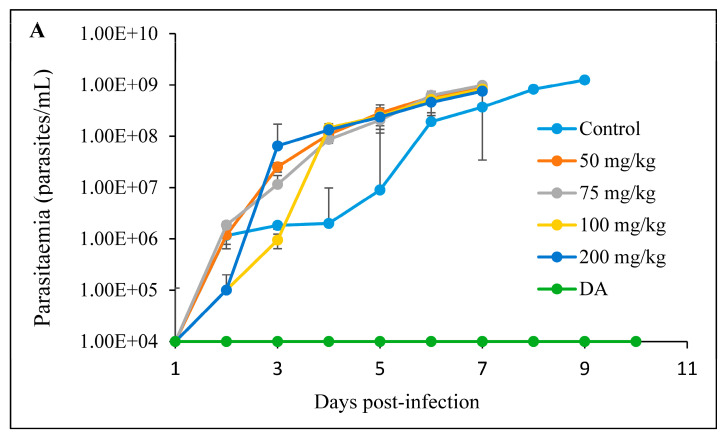
The evaluation of (**A**) parasitemia; (**B**) survival rate of the mice infected with *T. congolense* and intraperitoneally treated with different concentrations of CUR for 7 days. The 1 × 10^4^ represents parasitemia below the detection levels.

**Figure 9 pathogens-12-01227-f009:**
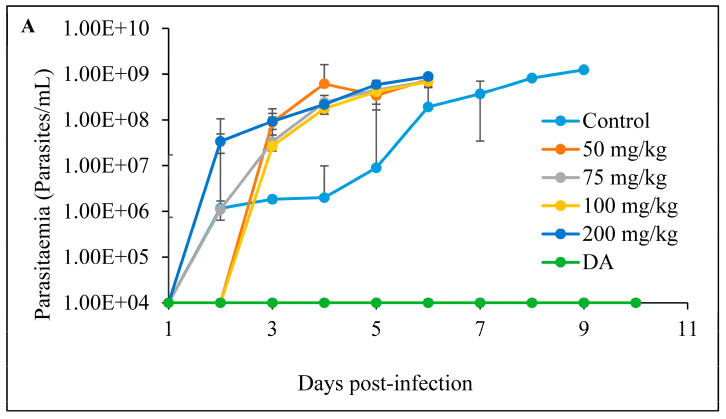
The evaluation of (**A**) parasitemia; (**B**) survival rate of the mice infected with *T. congolense* and intraperitoneally treated with different concentrations of CUR-nanoparticle. The 1 × 10^4^ represents parasitemia below the detection levels.

**Table 1 pathogens-12-01227-t001:** In vitro summary of the efficacy of CUR and CUR-nanoparticles against *T. congolense*, *T. b. brucei*, and *T. evansi* and the cytotoxicity on the MDBK and NIH 3T3 cells.

Test Samples	Anti-Trypanosomal Tests (µM ± S.D.)			Cytotoxicity Tests (µM ± S.D.)		Selectivity Index		
	*T. congolense*	*T. b. brucei*	*T. evansi*	MDBK	NIH 3T3	T. c	T. b. b	T. e
CUR	3.67 ± 0.31	7.61 ± 1.22	6.40 ± 3.07	50.51 ± 1.97	10.96 ± 3.09	8.37	4.01	4.80
ASPS	1.56 ± 0.50	28.16 ± 9.43	13.12 ± 0.13	54.27 ± 0.3	78.87 ± 1.7	42.67	2.36	5.07
WM	1.46 ± 0.49	46.25 ± 2.45	13.18 ± 0.81	57.21 ± 1.082	72.68 ± 0.68	44.49	1.40	4.93
EPN	1.70 ± 0.54	44.49 ± 8.1	39.96 ± 14.43	56.91 ± 0.89	56.97 ± 2.17	33.49	1.28	1.43

ASPS—Antisolvent precipitation with a syringe pump, WM—Wet milling method, EPN—Evaporative precipitation of nanosuspension. T. c: *T. congolense*; T. b. b: *T. b. brucei*; T. e: *T. evansi.*

## Data Availability

The compiled data for this study is linked directly to the figures and can be accessed at any point.

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
