# Peer review of "In Vivo Efficacy of Curcumin and Curcumin Nanoparticle in Trypanosoma congolense, Broden 1904 (Kinetoplastea: Trypanosomatidae)-Infected Mice"

_pathogens, 2023, doi:10.3390/pathogens12101227_

Round 1
Reviewer 1 Report
In the introduction of the paper entitled ‘In vivo efficacy of curcumin and curcumin nanoparticle in Trypanosoma congolense- infected mice’ the authors state that ‘Currently available trypanocides were developed decades ago and therefore, the development of alternative trypanocidal drugs remains important’. That statement is not correct, as shown in this paper by the 100% efficacy of DA for the treatment of AT. It does not matter how long ago they were developed, as long as they are effective. Moreover, the efficacy of CUR for AAT treatment, is shown to be ‘0’ which by itself is a proof that ‘timing’ is not an issue here.
What is lacking in the introduction is a proper discussion of available AAT treatment modalities, the problems that might occur, and recent developments in promising new drug treatments.
Result/discussion section: Overall, while the experiments are clearly explained and results are correctly presented, the overall paper is a large overinterpreted of the data and the authors should add to the title that CUR treatment has no curative effect on AAT. As such, it is not a candidate for further drug development, even not in its nanoparticle formulation. The effect observed in vitro is by no means a promising result when it comes to future perspectives as there are countless chemicals that will kill trypanosomes in vitro and for obvious pharmacological reasons can never be developed for in vivo use.
Technical issue: It seems in the in vivo approach there is a control missing, being control that can exclude the ‘particle component’ itself.
Scientific issue: this paper would be significantly strengthened if it would include some experiments that could start to explain the prolongation of survival after treatment. Inflammation in general is known to both positively contribute to parasitemia control, but negatively affect host survival, so basic experiments measuring inflammatory signal and effector molecules should be considered a minimal requirement to make this paper interesting for a wider reader public.
Discussion section: the last paragraph appears irrelevant. The authors argue about the possibility of BBB penetration of CUR to make it a candidate for HAT treatment. This makes no sense. If CUR is unable to have any curative effect in the models tested, where the BBB is not even an issue, it just shows that the approach is never going to work and cannot be taken forward in any credible way for the treatment of AAT or HAT for that matter. For HAT, recently 2 new drug compounds have been introduced that most likely will be able to eliminate HAT as a public health treat within the coming years. There is no reason to speculate on the future of CUR for the treatment of HAT.
Author Response
Good day sir/madam
Thank you for your thorough comments, please see the attachment.

Reviewer 2 Report
General comments:
In this study, the authors investigated the effects of Curcumin and CUR-nanoparticles on African trypanosomes (T. congolense, T. brucei, and T. evansi), as well as on mice infected with T. congolense. The study is well-designed, with a clear hypothesis and the experiments were executed according to a straightforward plan. CUR-nanoparticles showed important in-vitro and in-vivo effects in the close to micromolar range, with significant effects mostly observed on T. congolense. Furthermore, CUR-nanoparticles appear to better control parasitemia and improve survival rates in T. congolense-infected mice. Interestingly, no difference was observed in mice treated intraperitoneally with CUR alone or with CUR-nanoparticles. The manuscript is well-written and concise. However, there are certain parts that I considered less well-written, and could benefit from improvement and more information, as described below:
Methods:
The authors utilized a combination of methods to prepare curcumin nanoparticles under three main strategies. The sample size seems to be adequate, and the statistical analysis is clearly presented. There are no concerns about ethical requirements. Nevertheless, the description of the methods could be improved. Relevant information on the TEM instrument used, such as the type, brand, voltage, and camera type, was omitted. Similarly, for the cell culture, basic information such as the cultivation time, phase of the growth curve chosen to harvest the parasites, and whether serum was used or not, etc., are lacking. Similar issues were also found for mammalian cell lines. The same goes for the concentrations of CUR and CUR-nanoparticles used. Although one can infer from the figures, the exact concentration of each assay must be provided in the methods. The phrase "various concentrations" written in different parts of the text is not appropriate to describe what was done. Furthermore, information on the light microscope used for histopathology is also lacking.
Results
The analysis matches what was originally planned, and the results are clear. However, a few changes in figures and their corresponding descriptions are necessary to make the manuscript clearer. Minor essential revisions are suggested as follows:
Figure 1 does not include a scale bar, making it impossible to estimate the particle size. In the only TEM image provided, it is very difficult to observe and measure individual particles. It appears that there are significant size variations. The authors mention an average size of 120 nm, but what is the degree of monodispersity versus polydispersity? What is the standard deviation?
Selectivity versus Cytotoxicity:
It seems that the authors have averaged the cytotoxicity values for the two cell lines to calculate the selectivity index for each parasite. While this approach provides a more comprehensive view of the compounds' selectivity, it should be described more clearly in the methods and table description. Additionally, presenting the results separately for each cell line versus each parasite would offer more informative insights.
Histopathology
L- 294 Histological “sections”
Again, which compound was used in the experiments depicted in Figs 2 and 3? The authors simply refer to it as "CUR-nano treated mice."
Conclusion
The conclusions are well-supported by the data. The authors seem to have chosen simple approaches and well-planed experiments to address the main questions. The findings are thoroughly discussed, and their significance for chemotherapy against trypanosomes is evident.
Author Response
Good day sir/madam
Thank you for your comments, please see attachment.
Kind regards;

Reviewer 3 Report
This manuscript provides information about the "In vivo efficacy of curcumin and curcumin nanoparticles in Trypanosoma congolense- infected mice". It is an interesting research work and could contribute to the field. The manuscript is generally well-written except for the scientific names which must be written in full as Genus species, Author (Order: Family) in the main title and summary where they appear for the first time in the body text.
Author Response
Good day sir/madam
Thank you for your comments, please see the attachment.
Kind regards;

Round 2
Reviewer 1 Report
Thank you for the attempt to address the issues flagged during the first round of reviewing.
Author Response
Good day sir/madam
Thank you for your dedication to reviewing this paper. Your recommendations and suggestions are well received.
Kind regards;